# Simulation and Experiment of Manufacturing Process for Structural Aluminum Parts with Hard to Plastic Forming

**Chul Kyu Jin** 

School of Mechanical Engineering, Kyungnam University, 7 Kyungnamdaehak-ro, Masanhappo-gu, Changwon-si 51767, Korea; cool3243@kyungnam.ac.kr; Tel.: +82-55-249-2346; Fax: +82-505-999-2160

**Abstract:** A process comprising a hot extrusion process and a warm forging process was designed to form an umbrella-shaped aluminum structural component with a high degree of difficulty for the plastic forming method. A circular cylindrical part was extruded with a hot extrusion process, and then an embossing part was produced with a warm forging process. The formability and the maximum load required for forming were then determined using a forming analysis program. The hot extrusion process was executed at 450 °C under the extrusion speed at 6 mm/s, while the warm forging process was executed at 260 °C under the forging speed at 150 mm/s. The simulation results showed that the load required for hot extrusion was 1019 ton, while the load required for the warm forging was 534 ton. The umbrella-shaped part was manufactured by using a 1600 ton capacity press. The graphite lubricant was coated on the mold as well as the material. A forming experiment was performed under the same condition with the simulation condition. The portion where extrusion was done became elliptical with the $\alpha$-Al phase elongated towards extrusion direction. Whereas, the $\alpha$-Al phase became circular in the forged portion. The tensile strength value was found as 345 MPa, while elongation rate was 12%. Meanwhile, Vickers hardness value at the extruded portion was 105 HV, and it was 110 HV at the forged portion.

**Keywords:** hot extrusion; warm forging; simulation; aluminum; manufacturing process

## 1. Introduction

Metal parts with complicated forms are generally manufactured with a mechanical working (multi-axis machining) or casting process. With a mechanical process, though a product with a complicated shape can be manufactured, there is a drawback of longer process time resulting in mass production that is not possible with a high process cost. However, the casting process can make a part with complicated forms by implementing various casting methods such as die casting and sand casting depend on the price, shape, and mechanical properties of the product. In case of the parts for mechanical structure that require high mechanical properties, the part is manufactured with a sand casting process using ferrous iron or steel. Whereas, in case of the part (transport equipment) which is used in the mechanical structure and even requires light weight, it cannot use ferrous material with a high density. Therefore, a die casting process is used using non-ferrous materials such as aluminum or magnesium having a relatively low density. However, with a die casting method of aluminum or magnesium, it is almost not possible to manufacture a part having tensile strength more than 300 MPa. To manufacture a structural part for transport equipment, material should be lighter and should have a high mechanical strength. In terms of economic feasibility, material not only should have a low process cost but also its production should be faster. Therefore, with plastic working of aluminum alloy having a relatively low density and excellent formability, complex structural parts for transport equipment can be manufactured [1–3].

Aluminum alloy has various characteristics according to its type and is broadly categorized into casting alloy and structural alloy. Casting alloy refers to an alloy by increasing Si contents to improve fluidity and to reduce melting point, whereas structural alloy has an improved mechanical property by heat treatment after increasing Mg contents. As an aluminum alloy that can be implemented in structural parts, Al6061 has high Mg content. When alloy Al6061 is treated with T6 heat, due to precipitation of $Mg_2Si$, tensile strength can be improved to higher than 350 MPa [1,4].

In this study, the aim is to manufacture an umbrella-shaped part that is applied in the structure of transport equipment with a plastic working. However, this part cannot be manufactured with only a one-time plastic working. Therefore, a process design was executed to make the forming possible from two processes of hot extrusion and warm forging. Process design and die design drawing on experience from the workers at the site would not only increase process development time, but also could increase process cost and die cost due to trial and error [5–10]. For complex shapes, a simulation using a forming analysis program should be necessarily implemented. Then, a manufacturing process needs to be conducted from simulation data. The process design was validated using a forming analysis program and the forming load required for each process was calculated. The capacity of press equipment was chosen based on the forming load obtained by the simulation results. An experiment was performed using the boundary condition that was applied in the simulation. After manufacture the parts, their microstructures were analyzed and mechanical properties were measured.

## 2. Materials and Process Design for Plastic Working

### 2.1. Design of Plastic Working

Schematics for an aluminum part with an umbrella shape is presented in Figure 1. Its shape difficulty is considerably high for manufacturing with plastic working. A 3D design was conducted using a modeling program. The bottom of the part was a circular cylinder with a height of 57 mm and diameter of around Ø41.5 mm. The opposite side of the circular cylinder was rectangular (cut face) with an embossing having a height of 49 mm. Design was made in such way that a gradient of 3° at rectangular part and 1° at the circular cylinder were provided so that the part could be released easily from the die after forming. The volume of the part (cube size: 1.398 mm) measured from the modeling program was 357,508 mm$^3$. The weight calculated by applying density of aluminum 2.3 g/cm$^3$ was approximately 965 g.

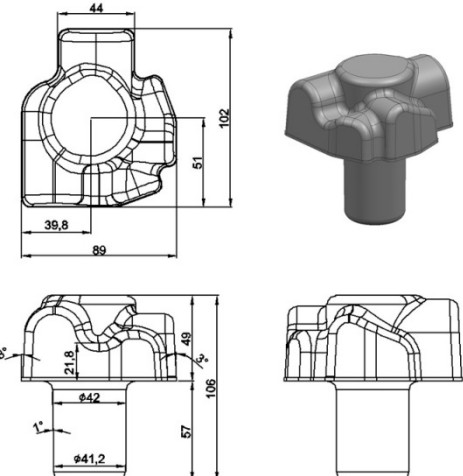

**Figure 1.** Drawing of the umbrella-shaped part.

Since volumes in situations of plastic working before deformation and after deformation are the same by bulk constant, when the circular cylinder part is manufactured with a die forging process, large amount of flash would be generated. In the manufacture of parts where flash is generated, a flow line

would be formed which causes irregularity or impairment of mechanical properties. Because flash is removed in the post-process, if it is large, a lot of time will be consumed. In order to minimize amount of flash generated, a circular cylinder part is extruded instead of forging. Since the extrusion ratio is approximately 4.1, it is expected that the extrusion load would be considerably large. If extrusion load is large, it becomes critical for die life. Therefore, work hardening effects should be avoided by implementing a hot extrusion. Furthermore, flow stress should be also minimized. In addition, flow stress value is abruptly changed according to the strain rate under hot temperature, thus extrusion speed should be slowed down to minimize the flow stress [7–12].

The opposite side of the circular cylinder can be produced as an embossing by the compression load from the die. Therefore, a warm die forging process was adopted [13–15]. The forming process for manufacture of an umbrella-shaped part is presented in Figure 2. A cylindrical part was extruded via hot extrusion as an initial specimen, and then embossing of the part at the opposite of the circular cylinder was produced by a warm forging. The volume of the initial specimen was 432,672 mm$^3$, and the cross-cut area was 5808 mm$^2$. The extrusion ratio was around 4.1, and the cylindrical column having a length of 57 mm was extruded. The shape manufactured by the extrusion was set as a specimen for forging process and was called as a blocker.

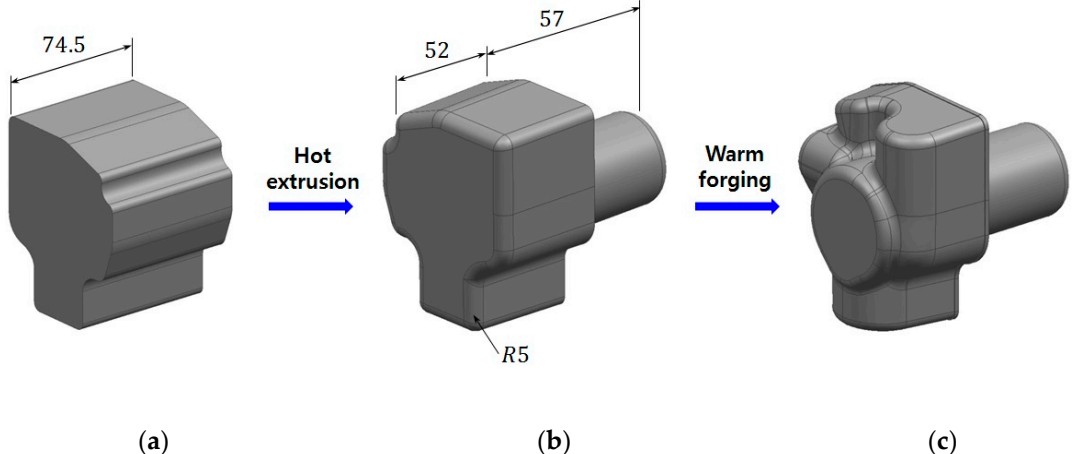

(**a**)　　　　　　　　　　　　　　(**b**)　　　　　　　　　　　　　　(**c**)

**Figure 2.** Forming process for fabricated umbrella-shaped part: (**a**) initial specimen; (**b**) blocker; (**c**) final part.

*2.2. Materials and Methods*

2.2.1. Hot Extrusion

Since the extrusion ratio was approximately 4.1, it was expected that the extrusion load would be considerably large. Therefore, a hot extrusion process was implemented at 450 °C which is almost 70% of the melting point of Al6061 (660 °C). SKD 61 as hot tool steel was used to die material.

The strain rate is defined as a speed at which plastic deformation is a flow stress function. Generally, flow stress of the material is slightly increased as strain rate increases at room temperature. However, under elevated temperatures, as strain rate increases, flow stress is significantly increased. That is, the extrusion process would only be smooth when the extrusion speed is set as slow as possible under elevated temperature conditions. Hence, the extrusion speed was set at 6 mm/s. The plasticity curve of true stress–true strain for strain rate 0.1 s$^{-1}$ and 10 s$^{-1}$ at 450 °C is showed in Figure 3. Data of true stress–true strain in Figure 3 obtained by tensile test at 450 °C at 0.1 strain rate and 10 strain rate, respectively. The diameter and gage length of specimen is 14 mm and 50 mm, respectively. The crosshead speed of tensile test machine is 5 mm/s at 0.1 strain rate and 500 mm/s at 10 strain rate, respectively.

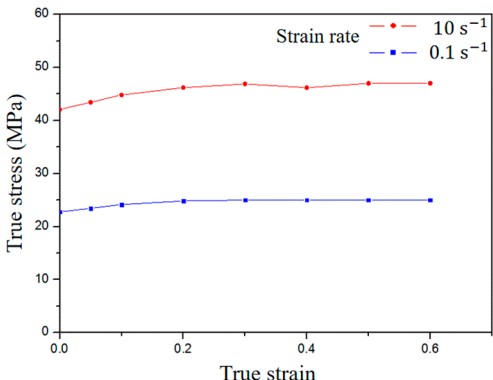

**Figure 3.** Flow curves of Al6061 at 450 °C for strain rate 0.1 s$^{-1}$ and 10 s$^{-1}$.

### 2.2.2. Warm Forging

A warm forging at 260 °C which is almost 40% of melting point of Al6061 material was implemented. Since strain rate is increased while embossing is produced, forging speed was set as low at 150 mm/s to minimize increases of flow strain by strain rate while forging. The initial height of the blocker with embossing produced was 52 mm, and forging speed was set at 150 mm, thus the initial strain rate was 2.88 s$^{-1}$. The height of the deepest embossing was 21.8 mm, and the strain rate was 6.88 s$^{-1}$. It was predicted that the strain rate while embossing was produced might be around 2.5~7.5. The plasticity curve of true stress–true strain for strain rate 0.1 s$^{-1}$ and 10 s$^{-1}$ at 260 °C is showed in Figure 4. Data of true stress–true strain in Figure 4 was obtained by tensile test at 260 °C at 0.1 strain rate and 10 strain rate, respectively.

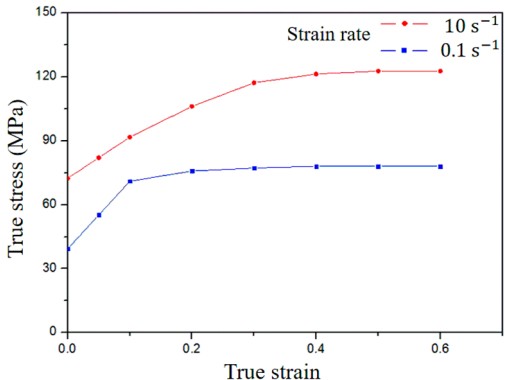

**Figure 4.** Flow curves of Al6061 at 260 °C for strain rate 0.1 s$^{-1}$ and 10 s$^{-1}$.

### 2.3. Measurement of Microstructures and Mechanical Properties

The specimens were collected from the extruded part and forged part to observe microstructure. The surface of the specimen was ground using a polishing equipment. Fine grinding was then performed using a polishing powder for the ground specimen by the emery paper. After that, the specimen was put into the etching solution for etching. Etching solution was prepared by diluting 190 mL of water, 5 mL of nitric acid, 3 mL of hydrogen chloride, and 2 mL of hydrofluoric acid. The microstructure was measured for the etched specimen using a light microscope.

The tensile test and Vickers hardness test were carried out to determine mechanical properties of the specimens. The specimens for tensile test were prepared as two sets according to specifications in ASTM E8M. The location of specimen collected for tensile test and Vickers hardness and the dimensions of the specimens are showed in Figure 5. The tensile specimens were collected from the direction of circular cylinder extrusion. The Vickers hardness was measured for the circular cylinder part (⑤) and four places of embossing (①, ②, ③, and ④). The gage length of the specimen was 36 mm, and

diameter of the specimen was Ø9 mm. For the tensile strength, specimens were collected from six numbers of the umbrella-shaped parts one from each and a total of six experiments were conducted. Vickers hardness was measured as 200 gf. Vickers hardness was measured five times at each position.

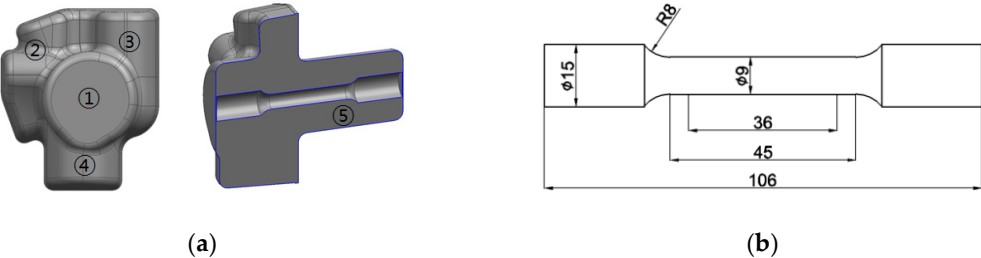

| (**a**) | (**b**) |

**Figure 5.** Position and dimension of tensile specimen and position of Vickers hardness: (**a**) Position of measurement (①~⑤: for Vickers hardness); (**b**) Dimension of tensile specimen.

## 3. Simulation for Plastic Working

### 3.1. Simulation Condition

#### 3.1.1. Hot Extrusion

Forming analysis was performed by ABAQUS software. Finite meshes were created for the die and the initial specimen to perform the finite element method. For the initial specimen, mesh generation was performed into three parts. The size of the mesh per each part was set differently each other. The mesh generation of initial specimen is showed in Figure 6. Because upper part of initial specimen is performed by extrusion process where deformation was large, mesh size was set at 0.3 mm. Upper part became the mesh densest among the part. Whereas, the bottom part where curvature is produced, the size of the mesh was set at 1 mm. At the intermediate part between upper part and bottom part, the size of the mesh was set at 9.5 mm to decrease calculation time. The total number of the mesh for the initial specimen generated was around 1,800,000.

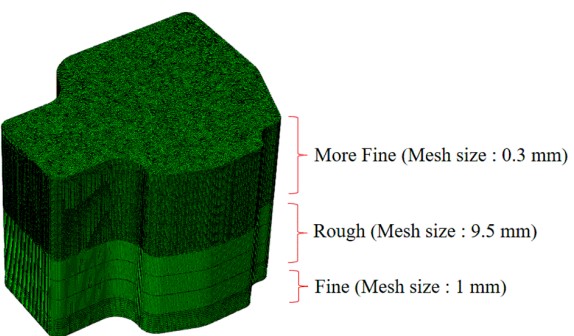

**Figure 6.** Mesh generation of initial specimen for extrusion process (blocker forming).

The materials properties for hot extrusion were input into the simulation program. Forging or extrusion are processes which generate large strains. The strain rate for the elastic strain is very small as compared to that of plastic strain. Therefore, analysis for elastic strain is executed with a strong plasticity type material wherein we can ignore elastic strain. Since simulation was done with a strong plasticity material condition, as can be seen in Figure 3, only the values in the plasticity area excluding elastic zone were applied. True stress–true strain data for the Al6061 aluminum alloy were used for the specimen. It was expected that the strain rate of extruded circular cylinder would be very large than other parts. Therefore, values from the true stress–true strain plasticity curve for strain rate $10 \text{ s}^{-1}$ and the plasticity curve value of true stress–true strain plasticity curve value for strain rate $0.1 \text{ s}^{-1}$ were applied. If the strain rate during forming analysis becomes in between $0.1 \text{ s}^{-1}$ and $10 \text{ s}^{-1}$,

the interpolated value from two true stress–true strain curve was adopted. Since, differences in strain rate are large with parts with large deformation and parts with small deformation, the true stress–true strain curve for the strain rate should be applied separately.

It was assumed that the used die was a steel body without deformation. The temperature increase caused by heat generated during plastic deformation was ignored by assuming heat generated by plastic deformation would be lost to outside. That means, the temperature of the material was assumed to be maintained at 450 °C while forming was underway. An elastic modulus 30 GPa of Al6061 alloy at 450 °C was applied and a Poisson's ratio of 0.3 (elasticity) was also applied.

Boundary conditions for forming a blocker by hot extrusion process is showed in Figure 7. Process was executed while making the die for forging static, and specimen was compressed while extrusion die was moving. Temperature of the die and material was set at 450 °C and this temperature was maintained while forming was executed. To extrude a circular cylinder part the extrusion die moves by 6 mm/s. A friction of 0.3 was applied between the die and material.

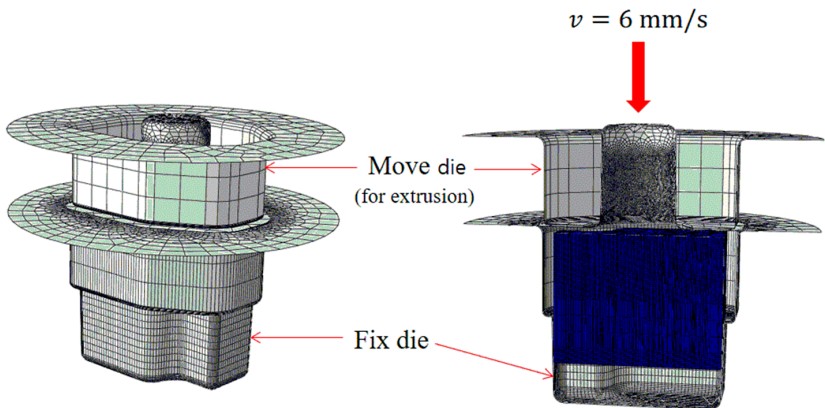

**Figure 7.** Boundary conditions of hot extrusion for blocker forming.

### 3.1.2. Warm Forging

Meshes were created from the blocker which was produced by hot extrusion process analysis. The created meshes on the blocker are showed in Figure 8. Since the opposite part of the circular cylinder was produced by forging process, smaller meshes in this part was generated than other parts. The size of mesh at bottom part for embossing forming was set at 1 mm, while the size of mesh in other parts was set at 3 mm. The total number of meshes created was around 340,000.

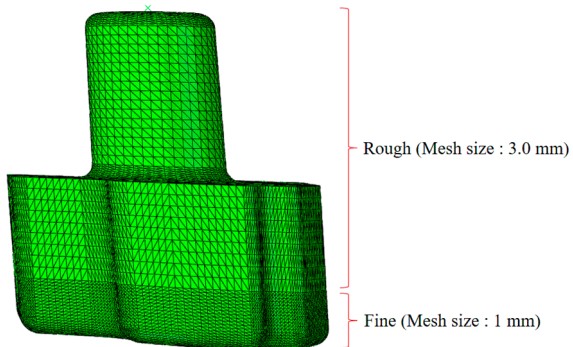

**Figure 8.** Mesh generation of blocker for forging process (embossing forming).

The materials properties for warm forging were input into the simulation program. The true stress–true strain curve for the strain rate 0.1 $s^{-1}$ for Al6061 and true stress–true strain curve for strain rate 10 $s^{-1}$ at 260 °C in Figure 4 were applied. If strain rate was in between 0.1 and 10 during forming, the interpolated data for two curves were applied. It was assumed that elastic modulus of Al6061 at

260 °C would be similar as in room temperature, therefore a value of 69 GPa was applied. In addition, the Poisson's ratio of 0.3 (elasticity) was adopted.

Boundary conditions to form the embossing part via warm forging is presented in Figure 9. Embossing die was fixed, while extrusion die moved, blocker was compressed inside of embossing die. The temperature of the die and material were set at 260 °C. While forming was continued, it was assumed that the temperature of the material was maintained to 260 °C. The extrusion die moved at a speed of 150 mm/s and compressed the blocker into the cavity of embossing die. A friction of 0.3 was applied between the die and material.

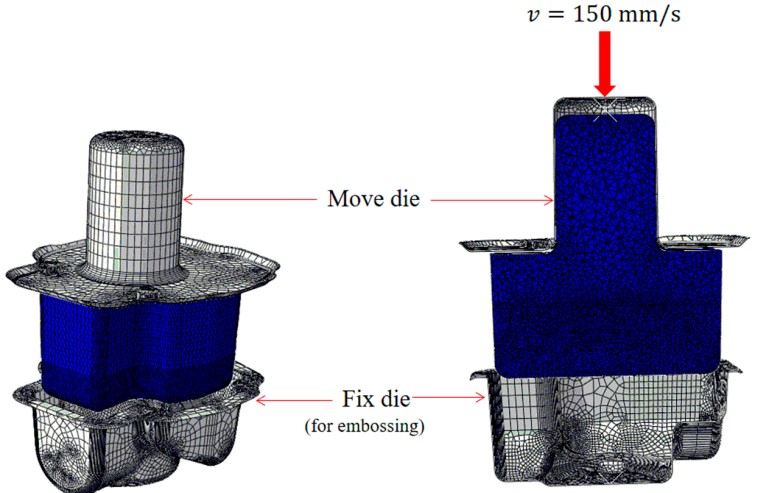

**Figure 9.** Boundary conditions of warm forging for embossing forming.

### 3.2. Simulation Results

### 3.2.1. Hot Extrusion

Simulation result for forming into blocker is showed in Figure 10. Since the extruded circular cylinder part had a large strain, shape of the element meshes was not smooth. While material was extruded into the circular cylinder part, it was expected that flow lines would also be formed near cross-cut of circular cylinder and rectangular shapes in the produced part. Whereas, at the end of circular cylinder where extrusion was completed, a band was produced. The band became a flash while going through forging, and was removed by trimming process.

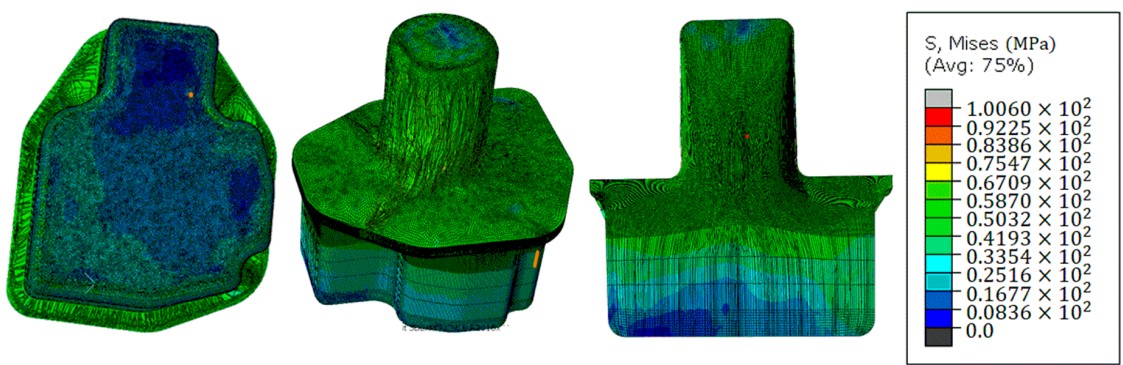

**Figure 10.** Simulation result of hot extrusion.

The graph of necessary load and displacement values drawn from the simulation results when the blocker was produced is showed in Figure 11. The capacity of the press to be used in the forming experiment can be determined from the load which was required for forming obtained from the

simulation results. As shown in the graph, if the displacement became 3 mm, the die would be contacting the specimen, and then compression could be executed. When the displacement was 5.5 mm, curvature was produced and circular cylinder was gradually extruded. The load imposed in the process was 1932 kN. The forming load was dramatically increased when circular cylinder part started forming. Whilst the displacement value became 7.0 mm, a circular cylinder part was completely produced. The load required when a circular cylinder was completely produced was 10,000 kN. To form a blocker from these simulation results, a press capacity of more than 1019 ton was required. The time to complete forming the blocker would be 7.5 s.

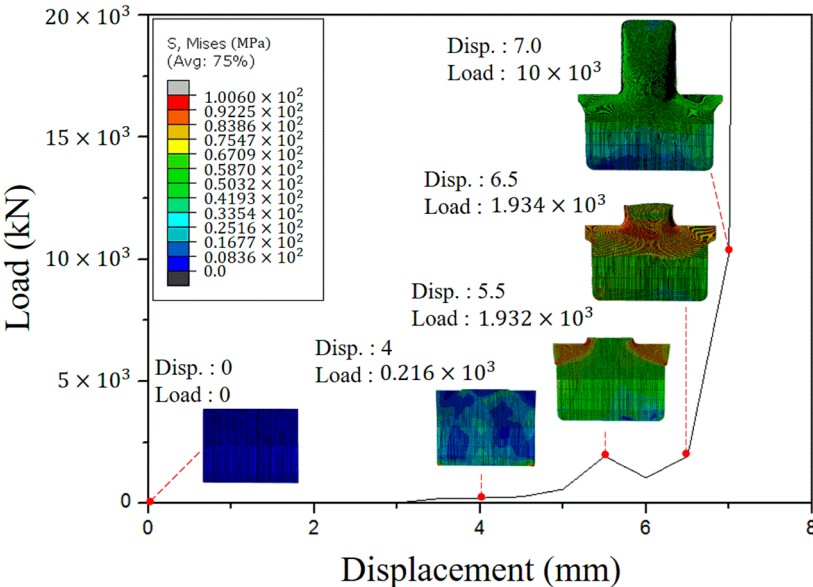

**Figure 11.** Load–displacement curve obtained by simulation result of hot extrusion.

### 3.2.2. Warm Forging

The simulation result for the embossing at the opposite for the circular cylinder is presented in Figure 12. It was the simulation results wherein forming was completed to make a final product having an umbrella shape. The embossing part was perfectly produced as in the modeling of umbrella shaped part in Figure 1. The band produced in the blocker was even greater while going through embossing forming. This phenomenon explains that flash is overflowing in actual forming process. Therefore, large amount of flash was expected in actual forming.

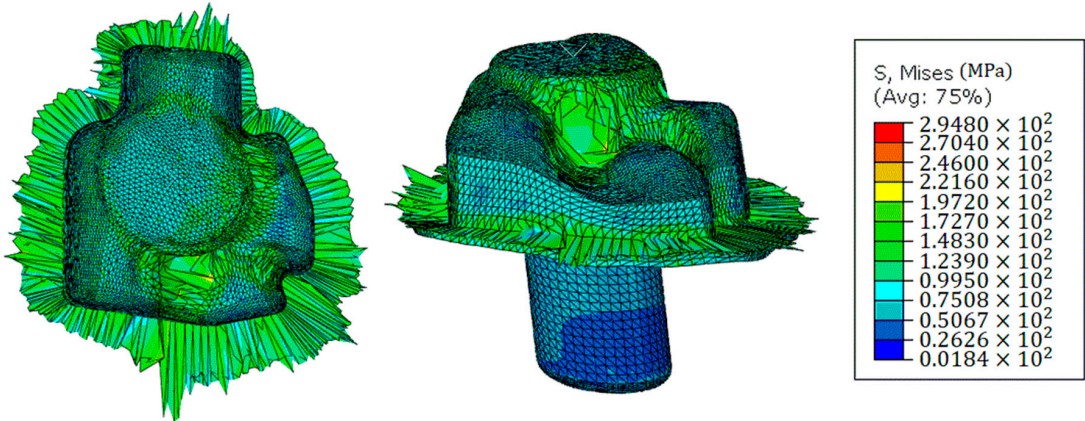

**Figure 12.** Simulation result of warm forging.

The graph of necessary load and displacement values drawn from the simulation results when the embossing part was produced is showed in Figure 13. It was confirmed from the graph that when the displacement value was 2.4 mm, the moving die contacted with the blocker and the embossing part started forming. When the displacement value was 3.0 mm, generally forming was executed at the right side with required load of 427 kN. When the displacement value became 3.37 mm, embossing part was completely produced. The required load when the embossing part was completely produced was 5237 kN. It was clear from the simulation result that a press having a capacity of at least more than 534 ton would be needed to form the embossing part. The time to form embossing completely would be around 0.33 s.

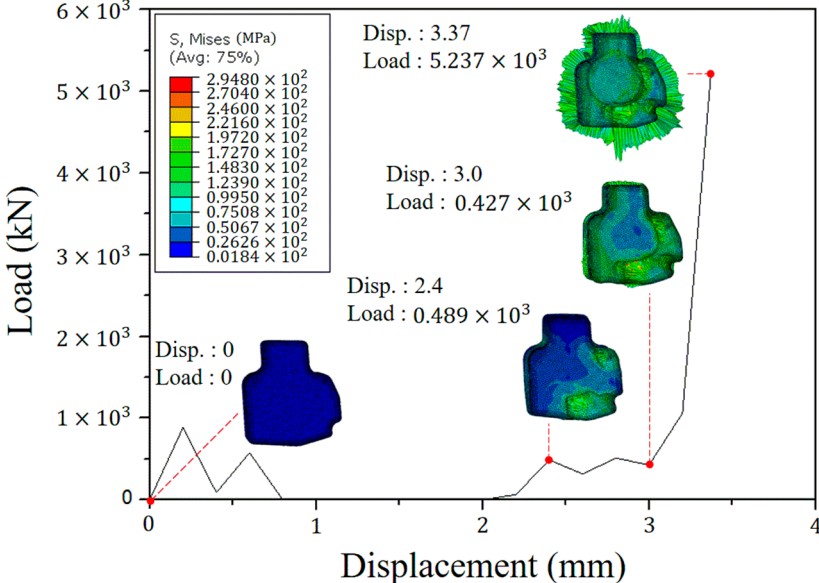

**Figure 13.** Load–displacement curve obtained by simulation result of warm forging.

## 4. Experiment for Plastic Working

### 4.1. Experimental Condition

The die for forming the blocker and embossing is presented in Figure 14a. Since a forming load of at least 1019 ton is required to form a blocker from the simulation results, a press capacity of 1600 ton was used in the actual experiment. The die was installed on the 1600 ton press to manufacture the umbrella-shaped part. The initial specimen to be produced into a blocker by extrusion is showed in Figure 14b. The initial specimen and die were heated to 450 °C same as simulation condition. The initial specimen was loaded into the electric furnace and a heater was installed in the die for preheating. The temperature of the initial specimen and die became stabilized to 450 °C from the surface until the core, the graphite lubricant (friction coefficient 0.3) was applied on the cavity of the die (for blocker). Since die temperature was high, the graphite lubricant was instantly burned and volatilized. Therefore, as soon as lubricant was coated, the initial specimen was placed in the cavity of the die. The descending speed of the press was set as fast until extrusion die touched the specimen. After extrusion die contacted to the specimen, the press speed was set at 6 mm/s. While the press descended at that speed, the circular cylinder was extruded from the initial specimen to form into a blocker. Since the press operated at 6 mm per second, the time consumed for extruding the circular cylinder was around 7 s, in accord with the simulation result.

To form the embossing part on the prepared blocker, blocker and die were cooled by air to reduce temperature to 260 °C. Once temperature of the surface and core became 260 °C, the graphite lubricant was coated on the cavity of the die. Then, blocker was inserted into the embossing cavity of the die and made the press descended fast so that die could be contacted to the surface of the extrusion die.

The press moved at 150 mm per second, compressing the blocker so that the embossing part was produced. When the press moved by 150 mm per second, the time required for forming the embossing part was 0.33, the same was also true with the analysis time.

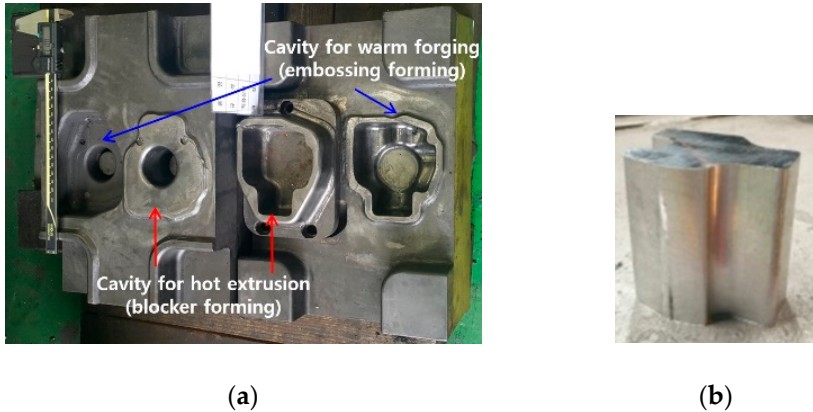

(**a**)　　　　　　　　　　　　　　　　　　　(**b**)

**Figure 14.** Tool and material for experiment. (**a**) Die of hot extrusion and warm forging. (**b**) Initial specimen.

*4.2. Experimental Results*

4.2.1. Forming Experiment

The blocker produced by the extrusion is showed in Figure 15a. The simulation results for the same angle are presented in Figure 15b. From the experiment and simulation, appearance of the specimens from two approach is almost same. In the blocker manufactured with the same as the simulation results, a band was produced.

**Experiment**　　　　　　　　　　　　　　　**Simulation**

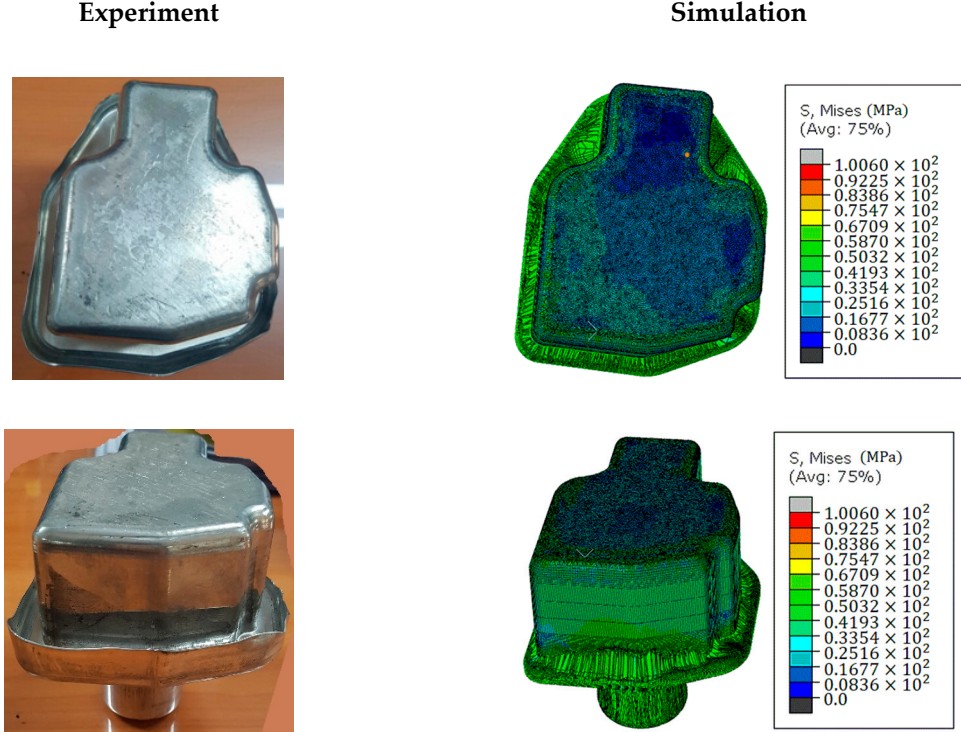

**Figure 15.** *Cont.*

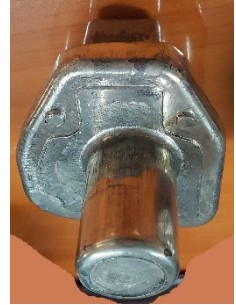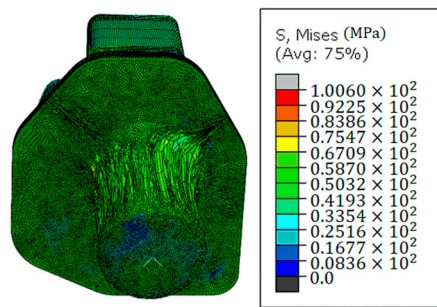

**Figure 15.** Experiment result of hot extrusion (produced blocker).

The maximum load measured from the load cell that was installed on the press was 1210 ton.

The final product on which embossing was produced on the blocker by the warm forging is presented in Figure 16a. The simulation results obtained from the same angle is showed in Figure 16b. From the experiment and simulation, appearance of the specimens from two approach is almost same. More flash was produced in the experiment than that in the simulation. The experimental results could vary a little in each experiment according to working condition. That means that the results can differ to some extent according to lubricant coating, temperature of die, and material. Therefore, the shape of the flash generated from the circular cylinder and embossing would be changed in each experiment.

**Experiment**　　　　　　　　　　　　　　　　　　**Simulation**

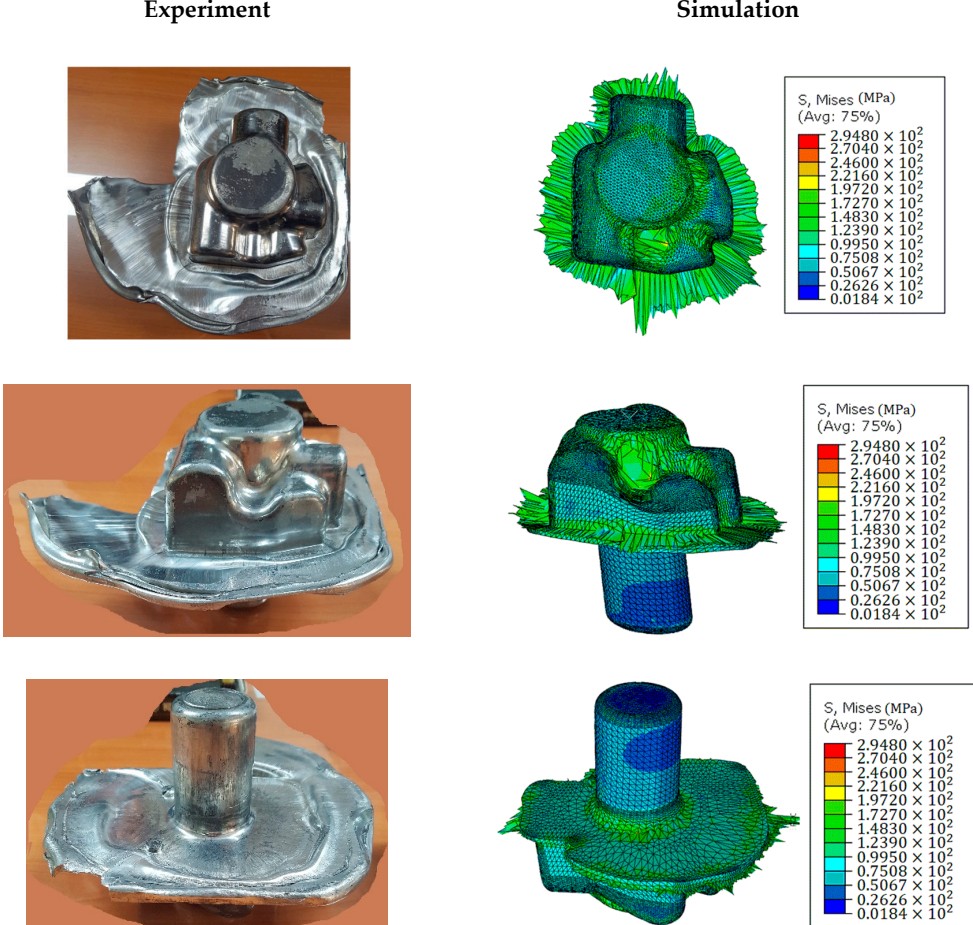

**Figure 16.** Experiment result of warm forging (produced final part).

The maximum load measured from the load cell was 600 ton which was 66 ton higher than that of the simulation results.

4.2.2. Microstructures and Mechanical Properties

The microstructure of the extruded part and forged part of the manufactured umbrella-shaped part is showed in Figure 17. The specimens were collected from extruded part and forged part. The microstructures at two locations clearly showed differences. The extruded part had an elliptical shape with long α-Al phase and the grain boundary was very narrow. The reason why the shape of the α-Al phase was elliptical was that there was a big strain towards extrusion direction when the circular cylinder part was extruded. Therefore, shape of α-Al phase was elongated towards extrusion direction inside the part also. The narrow grain boundary was attributed by a large extrusion ratio. When extrusion proceeded, the α-Al phase passed the narrow cut area of the circular cylinder, adhered to each other, making the boundary narrower. The microstructure of the forged part shows a round α-Al phase and was smaller than that in the extruded part. Furthermore, the boundary surface between α-Al phase also wider as compared with those in the extruded part. It is because forging was executed at 260 °C, resulting large numbers of process phase than in the extrusion, making a wider particle boundary. The shape of the α-Al phase was deformed and size became smaller by the compressive force of the forging.

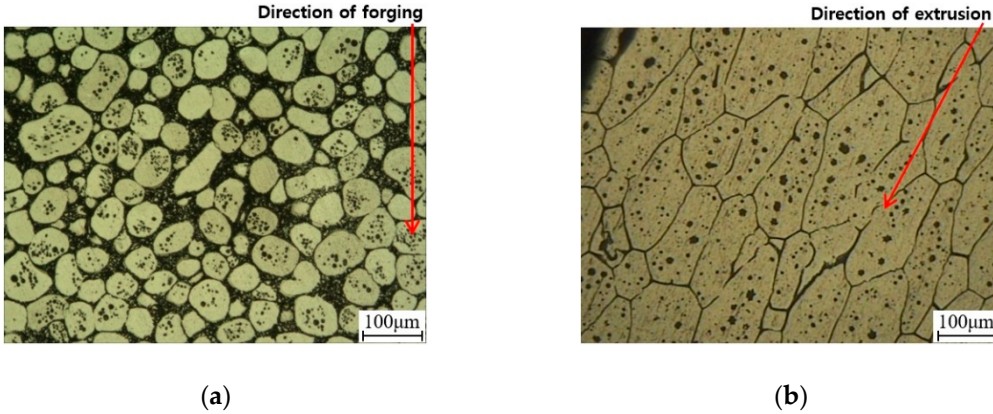

(**a**)                    (**b**)

**Figure 17.** Microstructures of produced final part: (**a**) forged part; (**b**) extruded part.

Table 1 shows results of tensile test and Vickers hardness. The yield strength was 290 MPa, tensile strength was 340 MPa, and elongation rate was 12%. Since deformation was occurred in a great extent while going through hot extrusion and warm forging, process hardening was also occurred greatly due to propagated dislocation. Vickers hardness showed above 110 HV in all forged part. While, the Vickers hardness at extruded part was 105 HV. Higher hardness was observed in the forged part than the in the extruded part. The reason can be found from the microstructures at two places in Figure 17. The higher hardness was also attributed by larger compressed part by forging than the circular part by extrusion. In general, the processed structure showed a higher hardness values than that of α-Al phase.

**Table 1.** Mechanical properties of produced final part.

| Yield Strength | Tensile Strength | | Elongation | | |
|---|---|---|---|---|---|
| 290 ± 6 MPa | 340 ± 8 MPa | | 12 ± 2% | | |
| **Position** | ① | ② | ③ | ④ | ⑤ |
| **Vicker's hardness (HV)** | 110 ± 3 | 112 ± 4 | 110 ± 3 | 111 ± 3 | 105 ± 2 |

## 5. Conclusions

A process comprising a hot forging and warm forging was designed to manufacture an umbrella-shaped part with a high degree of shape difficulty. The process design was validated using a forming

analysis program and the forming load required in each process was drawn. The actual forming experiment was executed using the simulation results.

(1) A blocker produced by applying the extrusion condition of hot extrusion at 450 °C under the extrusion speed at 6 mm. The forming load required by simulation results was found to be 1019 ton. While, the forming load was measured as 1210 ton in the experiment.

(2) The warm forging process for embossing forming was adopted at 260 °C under the speed at 150 mm/s. The forming load required by simulation results was found to be 534 ton as compared with the load 600 ton in the experiment. From the experiment and simulation results, appearance from the two approaches is almost the same.

(3) The microstructure of the produced part showed big differences in the extruded portion and forged portion. The extruded portion was elliptical where the $\alpha$-Al phase were elongated towards extrusion direction, while $\alpha$-Al phase were nearer to circle in the forged portion due to shape deformation by the compression.

(4) The tensile strength and elongation were 345 MPa and 12%, respectively. The Vickers hardness at the extruded portion was found as 105 HV, while it was above 110 HV in the forged part.

**Author Contributions:** C.K.J. conceived and designed the experiments; C.K.J. performed the simulation and experiments; C.K.J. analyzed the data; C.K.J. contributed reagents/materials/analysis tools; C.K.J. wrote the paper.

**Acknowledgments:** This work was supported by Kyungnam University Foundation Grant, 2018.

**Conflicts of Interest:** The author declares no conflict of interest.

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
