# Peer review of "Simulation and Experiment of Manufacturing Process for Structural Aluminum Parts with Hard to Plastic Forming"

_metals, doi:10.3390/met9020207_

Round 1

Reviewer 1 Report

The paper is well written.

My only concerns are the following:

The largest part of the references are more than 10 years old. Please add some more recent reference (e.g.

a) G. Angella, A Di Schino, R. Donnini, M. Richetta, C. Testani, A. Varone , AA7050 Al Alloy Hot.forging process for improved fracture toughness properties, Metals, 2019, 9, 64, doi: 10.3390/met9010064

b)  Maizza, G.; Pero, R.; Richetta, M.; Montanari, R. Continuous dynamic recrystallization (CDRX) model for aluminum alloys. J. Mater. Sci. 201853, 4563–4573.

Some concerns are about the quality of figures: In general they are too small and the legends difficult to be read.

Author Response

Comments and Suggestions for Authors

The paper is well written.

My only concerns are the following:

The largest part of the references are more than 10 years old. Please add some more recent reference (e.g.

a) G. Angella, A Di Schino, R. Donnini, M. Richetta, C. Testani, A. Varone , AA7050 Al Alloy Hot.forging process for improved fracture toughness properties, Metals, 2019, 9, 64, doi: 10.3390/met9010064

b)  Maizza, G.; Pero, R.; Richetta, M.; Montanari, R. Continuous dynamic recrystallization (CDRX) model for aluminum alloys. J. Mater. Sci. 201853, 4563–4573.

Answer : I am grateful to you for the valuable comments and suggestion. Two references were added in References Section, respectively.

Change :

3.       Angella, G.; Di Schino, A.; Donnini, R.; Richetta, M.; Testani, C.; Varone , A. AA7050 Al Alloy Hot.forging process for improved fracture toughness properties. Metals 2019, 9, 64.

4.        Maizza, G.; Pero, R.; Richetta, M.; Montanari, R. Continuous dynamic recrystallization (CDRX) model for aluminum alloys. J. Mater. Sci 2018, 53, 4563–4573.

Some concerns are about the quality of figures: In general they are too small and the legends difficult to be read.

Answer : I am grateful to you for the valuable comments and suggestion. Size of Figures 13,15,16 was enlarged and also legends in Figure 9,11,15,16 were enlarged to easily read.

Change : Please check revised manuscript

Reviewer 2 Report

The theme of the article is focused on the realization of a piece with a complex geometry for which it has proposed two different types of manufacturing approaches.

The article deals with a topic I think interesting for the industrial model also because the demonstration and feasibility are based on the comparison between simulation and experimentation, a comparison that seems to be quite successful. In fact, the simulation apparently managed to give a good indication about the processability of the proposed product.

In any case, however, in the current form, from my point of view, it needs a revision first of all from the structure of the article point of view. In fact, by reading the article, some technical-descriptive information is learned in different parts of the text while they should be given in the initial part, perhaps in a dedicated section such as “materials and methods”.An example is the introduction of section 2.1, 2.2.1 and 4.1.2.

Another missing element is an experimental plan that can best support the results of the simulation, both for a correct interpretation of the results of the simulation and for a probable reduction of the error between simulation and experimentation due to the variability of the process itself.

In addition, there are several errors of form in the English language and some in-depth analysis would be necessary. Following some suggestions that could improve the final quality of the paper:

Abstract – line 12: replace “by “with “for”;

line 57-58: the whole sentence should be reviewed in the English form; in general “before going for manufacture” and “an simulation…” are not correct, please review;

line 61: redundant use of forming, you can replace “ forming experiment” with “an experimentation campaign”; in general the whole sentence needs a review: “using forming condition that was applied in the simulation results” is unclear. You set some boundary conditions to obtain simulation results that should be compared. Please review;

Line 66-68: umbrella-shape is used too much. I think it should be reviewed also this definition because, in my opinion, It is an expression that creates misunderstanding. “is an similar umbrella” is incorrect, please review; in general, the first part of paragraph 2.1 should be reduced depending on the most interesting description of the object;

line 80: remove “to” in the sentence “should to be” ;

line 101-102: please review the form that is unclear and badly written;

line 103-107: please, try to explain better why you decide to divide into three different volumes the mesh density;

line 120-122: too many repetitions and please pay attention to “is increase” that is a bad error and mistake twice;

line 143: “forming…formed” please review;

line 147: “0.3 of friction factor…” can be replaced with “a friction of 0.3 was applied…”;

line 152: “the created meshes on the blocker is showed…” needs plural form  of the verb;

line 152-153: bad sentence structure and English form, please review;

line 160: why you used a Poisson’s ratio of 0.3?;

line 165: I don’t like “judge” when talking about manufacturing;

line 177: please read at the above suggestion of line 147;

Figure 9: the legend is too small, please increase its dimension and improve quality;

line 191-200: all this part needs to be reviewed in order to better understand the Figure 10 explanation. “…was contacted to die…”, “…started extruded.” and “…enabling complete the analysis” are bad English forms and need to be reviewed;

line 213-224: all this part is unclear due to an error in the x-axis of Figure 12. Check all references to the figure. “…required for forming the blocker, forming the embossing…” need to be reviewed;

line 255-260: all this part should be reported in a Materials and Methods dedicated section and better explained why this procedure is necessary. Rewrite the first sentence in “…from both extruded and forged part to observe microstructure.” “The was measured …” English form of the sentence is bad, please review;

line 276: please review the sentence “The blocker was formed to a shape almost same as in the simulation result”;

line 281 - 289: this observation actually refers to different parts in the text: I don’t like “was formed” as an english expression. Saying “More flash was formed in the experiment..:” is unclear, it is better to write produced instead of formed, or “we detected the presence of...”. At line 285 you say “the results differ….according to lubricant coating, temperature of die and material, and skill level of the worker”. Well, in my opinion, the skill level of a worker is a factor that should be analyzed. So, when you have doubt of correlation between response variable and factors affecting the process, you can build a DoE plan that should help you to understand, for example, if the variable “operator/worker” is affecting the results by using different operators a block design. So, please, don’t evaluate results without certainty of what you are saying. Experimental campaign to support simulation results should improve the work done a lot and should help to better understand, for example, why some simulation results are so far from experimental ones or better the variability of the process maybe can reduce this difference.

References: please review this part because, for example, citation 4 and 11 are the same

Author Response

Comments and Suggestions for Authors

The theme of the article is focused on the realization of a piece with a complex geometry for which it has proposed two different types of manufacturing approaches.

The article deals with a topic I think interesting for the industrial model also because the demonstration and feasibility are based on the comparison between simulation and experimentation, a comparison that seems to be quite successful. In fact, the simulation apparently managed to give a good indication about the processability of the proposed product.

Answer : I am grateful to you for the valuable comments and suggestion.

In any case, however, in the current form, from my point of view, it needs a revision first of all from the structure of the article point of view. In fact, by reading the article, some technical-descriptive information is learned in different parts of the text while they should be given in the initial part, perhaps in a dedicated section such as “materials and methods”.An example is the introduction of section 2.1, 2.2.1 and 4.1.2.

Answer : I am grateful to you for the valuable comments and suggestion. The section of Materials and methods was added. This section is consisting of 2.2.1 hot extrusion, 2.2.2 warm forging and 2.2.3 Measurement of Microstructures and Mechanical Properties.

Change : Composition of manuscript was changed as below

1. Introduction

2. Materials and Process Design for Plastic Working

2.1. Design of plastic working

2.2. Materials and Methods

2.2.1. Hot Extrusion

2.2.2. Warm Forging

2.3. Measurement of Microstructures and Mechanical Properties

3. Simulation for Plastic Working

3.1 Simulation Condition

3.1.1. Hot Extrusion

3.1.2. Warm Forging

3.2. Simulation Results

3.2.1. Hot Extrusion

3.2.2. Warm Forging

4. Experiment for Plastic Working

4.1. Experimental Condition

4.2. Experimental Results

4.2.1. Forming Experiment

4.2.2. Microstructures and Mechanical Properties

5. Conclusions

Another missing element is an experimental plan that can best support the results of the simulation, both for a correct interpretation of the results of the simulation and for a probable reduction of the  error between simulation and experimentation due to the variability of the process itself.

Answer : I agree with your comments. I am grateful to you for the valuable comments and suggestion.

In addition, there are several errors of form in the English language and some in-depth analysis would be necessary. Following some suggestions that could improve the final quality of the paper:

Answer : I am grateful to you for the valuable comments and suggestion. I have a little time for English language editing because I will have to submit revised manuscript by 3 February. So, I will take English language editing service from professional company after I submit revised manuscript.

Abstract – line 12: replace “by “with “for”;

Answer : I am grateful to you for the valuable comments and suggestion. I changed by to for

line 57-58: the whole sentence should be reviewed in the English form; in general “before going for manufacture” and “an simulation…” are not correct, please review;

Answer : I am grateful to you for the valuable comments and suggestion. I revised these sentences

Change :

In case of a part with a high shape difficulty, an simulation using a forming analysis program should be necessarily implemented. And then manufacturing process needs to be conducted from simulation data.

line 61: redundant use of forming, you can replace “ forming experiment” with “an experimentation campaign”; in general the whole sentence needs a review: “using forming condition that was applied in the simulation results” is unclear. You set some boundary conditions to obtain simulation results that should be compared. Please review;

Answer : I am grateful to you for the valuable comments and suggestion. I revised these sentences

Change :

The capacity of press equipment was chosen based on the forming load obtained by the simulation results. Experiment was performed using the boundary condition that was applied in the simulation.

Line 66-68: umbrella-shape is used too much. I think it should be reviewed also this definition because, in my opinion, It is an expression that creates misunderstanding. “is an similar umbrella” is incorrect, please review; in general, the first part of paragraph 2.1 should be reduced depending on the most interesting description of the object;

Answer : I am grateful to you for the valuable comments and suggestion. I revised these sentences

Change :

The drawing of structural aluminum part having an umbrella shape is showed in Figure 1. Its shape difficulty is considerably high for manufacture with plastic working. A 3D design was conducted using a modelling program.

line 80: remove “to” in the sentence “should to be” ;

Answer : I am grateful to you for the valuable comments and suggestion. I removed should to be in this sentence.

Change : Because flash is removed in the post-process,

line 101-102: please review the form that is unclear and badly written;

Answer : I am grateful to you for the valuable comments and suggestion. I revised these sentences

Change :

Finite meshes were created for the die and the initial specimen to perform the finite element method. For the initial specimen, mesh generation was performed into three parts. The size of the mesh per each part was set differently each other. The mesh generation of initial specimen is showed in Figure 6.

line 103-107: please, try to explain better why you decide to divide into three different volumes the mesh density;

Answer : I am grateful to you for the valuable comments and suggestion. I revised these sentences

Change :

Because upper part of initial specimen is performed by extrusion process where deformation was large, mesh size was set at 0.3 mm. Upper part became the mesh densest among the part. Whereas, the bottom part where curvature is formed, the size of the mesh was set at 1 mm. At the intermediate part between upper part and bottom part, the size of the mesh was set at 9.5 mm to decrease calculation time. The total number of the mesh for initial specimen generated was around 1,800,000.

line 120-122: too many repetitions and please pay attention to “is increase” that is a bad error and mistake twice;

Answer : I am grateful to you for the valuable comments and suggestion. I revised these sentences

Change :

Generally, flow stress of the material is slightly increased as strain rate increase at room temperature. However, under the elevated temperature, as strain rate increase, flow stress is significantly increased.

line 143: “forming…formed” please review;

Answer : I am grateful to you for the valuable comments and suggestion. I revised these sentences

Change :

Boundary conditions for forming a blocker by hot extrusion process is showed in Figure 7.

line 147: “0.3 of friction factor…” can be replaced with “a friction of 0.3 was applied…”;

Answer : I am grateful to you for the valuable comments and suggestion. I revised these sentences

Change :

A friction of 0.3 was applied between the die and material.

line 152: “the created meshes on the blocker is showed…” needs plural form  of the verb;

Answer : I am grateful to you for the valuable comments and suggestion. I revised these sentences

Change :

The created meshes on the blocker are showed in Figure 8.

line 152-153: bad sentence structure and English form, please review;

Answer : I am grateful to you for the valuable comments and suggestion. I revised these sentences

Change :

Since the opposite part of the circular cylinder was formed by forging process, smaller meshes in this part was generated than other parts. The size of mesh at bottom part for embossing forming was set at 1 mm, while the size of mesh in other parts was set at 3 mm.

line 160: why you used a Poisson’s ratio of 0.3?;

Answer : I am grateful to you for the valuable comments and suggestion. Posisson’s ratio of 0.3 is applied in elastic interval.

line 165: I don’t like “judge” when talking about manufacturing;

Answer : I am grateful to you for the valuable comments and suggestion. I replace judge with predict

Change :

It was predicted that the strain rate while embossing was formed might be around 2.5 ~ 7.5.

line 177: please read at the above suggestion of line 147;

Answer : I am grateful to you for the valuable comments and suggestion. I revised these sentences

Change :

A friction of 0.3 was applied between the die and material.

Figure 9: the legend is too small, please increase its dimension and improve quality;

Answer : We are grateful to you for the valuable comments and suggestion. Size of legends in Figure 9,11,15,16 were enlarged to easily read.

Change : Please check revised manuscript

line 191-200: all this part needs to be reviewed in order to better understand the Figure 10 explanation. “…was contacted to die…”, “…started extruded.” and “…enabling complete the analysis” are bad English forms and need to be reviewed;

Answer : I am grateful to you for the valuable comments and suggestion. I revised these sentences

Change :

The graph of necessary load and displacement values drawn from the simulation results when the blocker was formed is showed in Figure 11. The capacity of the press to be used in the forming experiment can be determined from the load which was required for forming obtained from the simulation results. As show in graph, If the displacement became 3 mm the move die was contacted to specimen. And then compression could be executed. When the displacement was 5.5 mm, curvature was formed and circular cylinder was gradually extruded. The load imposed in the process was 1.9 kN. The forming load was dramatically increased when circular cylinder part started forming. Whilst, the displacement value became 7.5 mm, circular cylinder part was completely formed. The load required when circular cylinder was completely formed was 10,000 kN. To form a blocker from this simulation results, a press capacity more than 1,019 ton was required. The time to complete forming the blocker would be 7.5 sec..

line 213-224: all this part is unclear due to an error in the x-axis of Figure 12. Check all references to the figure. “…required for forming the blocker, forming the embossing…” need to be reviewed;

Answer : I am grateful to you for the valuable comments and suggestion. I revised these sentences

Change :

The graph of necessary load and displacement values drawn from the simulation results when the embossing part was formed is showed in Figure 13. It was confirmed from the graph that when the displacement value was 2.4 mm, the move die was contacted to the blocker and embossing part was started forming. When the displacement value was 3.0 mm, generally forming was executed at the right side with required load of 427 kN. When the displacement value became 3.37 mm, embossing part was completely formed. The required load when the embossing part was completely formed was 5,237 kN. It was clear from the simulation result that a press having capacity at least more than 534 ton would be needed to form the embossing part. The time to form embossing completely would be around 0.33 sec.

line 255-260: all this part should be reported in a Materials and Methods dedicated section and better explained why this procedure is necessary. Rewrite the first sentence in “…from both extruded and forged part to observe microstructure.” “The was measured …” English form of the sentence is bad, please review;

Answer : I am grateful to you for the valuable comments and suggestion. The section of Materials and methods was added. This section is consisting of 2.2.1 hot extrusion, 2.2.2 warm forging and 2.2.3 Measurement of Microstructures and Mechanical Properties.

First sentences in section of Measurement of Microstructures and Mechanical Properties was revised.

Change :

The specimens were collected from the extruded part and forged part to observe microstructure. The surface of the specimen was ground using a polishing equipment. Fine grinding was then performed using a polishing powder for the ground specimen by the emery paper. After that, the specimen was put into the etching solution for etching. Etching solution was prepared by diluting 190 ml of water, 5 ml of nitric acid, 3 ml of hydrogen chloride, and 2 ml of hydrofluoric acid. The microstructure was measured for the etched specimen using an light microscope.

line 276: please review the sentence “The blocker was formed to a shape almost same as in the simulation result”;

Answer : I am grateful to you for the valuable comments and suggestion. I remove this sentence.

line 281 - 289: this observation actually refers to different parts in the text: I don’t like “was formed” as an english expression. Saying “More flash was formed in the experiment..:” is unclear, it is better to write produced instead of formed, or “we detected the presence of...”. At line 285 you say “the results differ….according to lubricant coating, temperature of die and material, and skill level of the worker”. Well, in my opinion, the skill level of a worker is a factor that should be analyzed. So, when you have doubt of correlation between response variable and factors affecting the process, you can build a DoE plan that should help you to understand, for example, if the variable “operator/worker” is affecting the results by using different operators a block design. So, please, don’t evaluate results without certainty of what you are saying. Experimental campaign to support simulation results should improve the work done a lot and should help to better understand, for example, why some simulation results are so far from experimental ones or better the variability of the process maybe can reduce this difference.

Answer : I am grateful to you for the valuable comments and suggestion. I revised these sentences. the skill level of a worker was removed in these sentence.

All of the expression “was formed” replace with “was produced”.

Change :

The final product on which embossing was formed on the blocker by the warm forging is showed in Figure 16 (a). The simulation results obtained from the same angle is showed in Figure 16 (b). From the experiment and simulation, appearance of the specimens from two approach is almost same. More flash was produced in the experiment than that in the simulation. The experimental results could vary a little in each experiment according to working condition. That means, the results can differ to some extent according to lubricant coating, temperature of die and material. Therefore, the shape of the flash generated from the circular cylinder and embossing would be changed in each experiment.

The maximum load measured from the load cell was 600 ton which was higher by 66 ton than that of simulation results.

References: please review this part because, for example, citation 4 and 11 are the same

Answer : I am grateful to you for the valuable comments and suggestion. Reference 4 was changed.

Change :

4.       Maizza, G.; Pero, R.; Richetta, M.; Montanari, R. Continuous dynamic recrystallization (CDRX) model for aluminum alloys. J. Mater. Sci 2018, 53, 4563–4573.

Round 2

Reviewer 2 Report

The authors have received positively a good part of the suggestions reported in the previous revision. Now the article seems a lot smoother and clearer. I confirm the good quality of the work done. I recommend, for the future, to deepen the comparison with the simulation by implementing an appropriate experimental plan with which to further clarify some of the differences emerged in this phase.